# Prospective, Randomized, Comparative Study of Myeloablative Fludarabine/Busulfan and Fludarabine/Busulfan/Total Body Irradiation Conditioning in Myeloid Diseases

**DOI:** 10.3390/cancers17071140

**Published:** 2025-03-28

**Authors:** Hyung C. Suh, Scott D. Rowley, Sukhdeep Kaur, Brittany Lukasik, Phyllis McKiernan, Michele Boonstra, Melissa Baker, Mary DiLorenzo, Alan Skarbnik, Jason Voss, Alexandra Hampson, Bianca DeAgresta, Brighid Boylan, Themba Nyirenda, David H. Vesole, Michele L. Donato

**Affiliations:** 1John Theurer Cancer Center, Hackensack Meridian Health, Hackensack, NJ 07601, USA; sdr62@georgetown.edu (S.D.R.); sukhdeep.kaur@hmhn.org (S.K.); brittany.lukasik@hmhn.org (B.L.); phyllis.mckiernan@hmhn.org (P.M.); michele.boonstra@hmhn.org (M.B.); melissa.baker@hmhn.org (M.B.); jason.voss@hmhn.org (J.V.); alexandra.dellapia@hmhn.org (A.H.); bianca.deagresta@hmhn.org (B.D.); david.vesole@hmhn.org (D.H.V.); michele.donato@hmhn.org (M.L.D.); 2MedStar Georgetown University Hospital, Washington, DC 20007, USA; 3Novant Health, Charlotte, NC 28204, USA; medilorenzo@novanthealth.org (M.D.); azskarbnik@novanthealth.org (A.S.); 4Hackensack Meridian School of Medicine, Hackensack, NJ 07601, USA; brighid.boylan@hmhn.org; 5Department of Biostatistics, Bioinformatics, and Biomathematics, Hackensack University Medical Center, Hackensack, NJ 07601, USA; themba.nyirenda@hmhn.org

**Keywords:** allogeneic hematopoietic stem cell transplantation, acute myeloid leukemia, myeloablative conditioning, fludarabine, busulfan, total body irradiation, prospective, randomized clinical trial

## Abstract

Allogeneic hematopoietic stem cell transplantation can cure myeloid diseases, such as acute myeloid leukemia (AML), but relapse after transplant can be fatal. Myeloablative conditioning regimens, such as fludarabine/busulfan (Flu/Bu), can help reduce the risk of relapse. Previous retrospective studies have shown that adding total body irradiation (TBI) to conditioning with Flu/Bu in acute leukemia can help reduce recurrence. This prospective, randomized study sought to confirm these findings by comparing two conditioning regimens—sequential 4-day regimen of Flu/Bu (Flu/Bu4) versus Flu/Bu4 and total body irradiation (400 cGy)—in patients with myeloid diseases, ~75% of whom had AML. In the AML cohort and holistic study population, risk of relapse was lower with Flu/Bu4/TBI than Flu/Bu4 but this was not statistically significant. Non-relapse mortality was increased in the Flu/Bu4/TBI arm. Consequently, overall survival at 1 year was similar between both treatment arms, meaning no survival advantage was observed with adding TBI to Flu/Bu4.

## 1. Introduction

Allogeneic hematopoietic stem cell transplantation (alloSCT) is a critical treatment modality in adults with myeloid malignancies. The anti-leukemic effects of alloSCT are attributed to the graft-versus-leukemia (GVL) effect mediated by donor cells and are also a direct result proportional to the intensity of the pre-transplant conditioning regimen. However, the benefits of myeloablative conditioning (MAC) are limited (especially for older patients and those with co-morbid illnesses) by considerable non-relapse mortality (NRM) secondary to treatment-related toxicity, graft-versus-host disease (GvHD), and infectious complications [1,2].

To overcome the higher risks of NRM with MAC, reduced-intensity conditioning (RIC) regimens for alloSCT were developed for use in various hematologic malignancies. The advent of RIC regimens permits the extension of a potentially curative GVL effect to patients in whom transplantation using MAC is prohibited by comorbidities or age [3,4,5]. At present, most allografts performed in the United States use an RIC regimen [6,7], though higher rates of relapse have been identified with the utilization of an RIC regimen when compared with MAC [8,9].

Studies comparing total body irradiation (TBI)-based MAC regimens identified dose-dependent effects with an inverse relationship between anti-leukemic activity and NRM [10,11,12]. Busulfan was developed as an alternative to TBI [13]. A phase I study initially demonstrated the maximum dose of oral busulfan as 16 mg/kg given concomitantly with cyclophosphamide (Cy) [14]. Subsequent studies demonstrated the variability of serum levels using the oral regimen [15], development and tolerability of an intravenous formulation [16], and decreased toxicity with substitution of fludarabine for Cy [17]. The lower toxicity of the fludarabine/busulfan (Flu/Bu) regimen presented the possibility of increasing the intensity of the regimen to achieve a lower risk of relapse.

A retrospective study published in 2010 by Russell et al. evaluated a sequential 4-day regimen of fludarabine/busulfan (Flu/Bu4) with or without the addition of 400 cGy of TBI for pre-transplant conditioning of patients with acute leukemia [18]. The researchers suggested, based on this non-randomized study, that adding 400 cGy TBI to Flu/Bu4 significantly reduced relapse and improved both overall survival (OS) and disease-free survival, with no differences in NRM observed at 3 years. However, the benefits of adding TBI to Flu/Bu4 have not been evaluated in a prospective, randomized trial. Therefore, this prospective randomized clinical trial was designed to confirm the retrospective study results.

## 2. Materials and Methods

This is a prospective, randomized clinical trial that enrolled 48 adult patients with myeloid diseases undergoing alloSCT from July 2010 to August 2019 at Hackensack University Medical Center (Figure 1). The eligible subjects for this study were those who had diagnosis of a myeloid malignancy and had a human leukocyte antigen (HLA)-compatible related or unrelated stem cell donor. Two subjects who did not receive stem cell transplantation were considered screen failures and are excluded from the analysis (final accrual n = 46). Study accrual was halted earlier than planned (target n = 52 subjects) when interim analysis showed that adding TBI to Flu/Bu4 resulted in increased regimen-related mortality.

Institutional Review Board (IRB) approval for this study was obtained from Hackensack Meridian Health’s IRB (Pro00001278), and written informed consent was obtained from all subjects. The study was conducted under the International Conference on Harmonization Good Clinical Practice Guidelines and according to the Declaration of Helsinki. 

### 2.1. Treatment

Subjects were randomized to Flu/Bu4 or Flu/Bu4/TBI conditioning prior to alloSCT. The preparative myeloablative chemotherapy included fludarabine 40 mg/m^2^ based on the following dosing convention: adjusted body weight if a patient weighs 120–150% of ideal body weight (IBW); actual body weight (ABW) if a patient weighs less than 100% of IBW. Fludarabine was administered intravenously once daily on days -6 to -3 followed by busulfan 130 mg/m^2^ intravenously once daily on days -6 to -3 (Flu/Bu4). Busulfan pharmacokinetic measurements were not used to adjust the busulfan serum levels in either cohort. The Flu/Bu4/TBI cohort also received 400 cGy TBI given in two 200 cGy fractions on day -1. All subjects received rabbit anti-thymocyte globulin (rATG; Thymoglobulin^®^) at doses of 0.5 mg/kg intravenously on day -3, 1.5 mg/kg intravenously on day -2, and 2 mg/kg intravenously on day -1.

Donors were chosen based on their compatibility for HLA-A, B, and DRB1 for related donors or HLA-A, B, C, and DRB1 for unrelated donors (URD). The source of hematopoietic stem cells (HSC) and donor age or sex, ABO compatibility, HLA match, and cytomegalovirus (CMV) status were not defined by protocol and were ascertained by the treating physician. In accordance with institutional practices, male sex and younger age were prioritized in donor selection. Collection of stem cell grafts was performed via standard techniques. Excluding cryopreservation for donor management and red blood cell depletion of major mismatched, ABO-incompatible bone marrow grafts, no graft processing was performed. Peripheral blood stem cell (PBSC) grafts were collected using standard techniques after granulocyte colony-stimulating factor (G-CSF) mobilization with a target dose of 4 to 6 × 10^8^ CD34+ cells/kg patient weight. The target threshold for bone marrow harvesting was >3 × 10^8^ nucleated cells per kilogram; however, cells were infused without adjustment regardless of quantity. The National Marrow Donor Program or similar registries were used to source URD grafts. The median number of CD34+ stem cells per kilogram of body weight was 4.02 × 10^6^ (range: 1.96–4.21 × 10^6^) and 2.33 × 10^6^ (range: 2.2–2.46 × 10^6^) in the PBSC and BM grafts, respectively. Day 0 was defined as the day when cell product infusion was completed. Forty subjects received PBSCs. Six subjects received bone marrow (BM) grafts.

All patients received an acute GvHD (aGvHD) prophylaxis regimen of tacrolimus, which was initiated on day -1, adjusted to a target therapeutic level of 5–15 ng/mL, and tapered off by day 180 in absence of GvHD, and methotrexate 5 mg/m^2^ (ABW) intravenously on days 1, 3, 6, and 11 post-transplantation. rATG was given as described above. Supportive care was administered as described in Appendix A.

Initial treatment for aGvHD included prednisone or methylprednisolone in combination with continued tacrolimus. First-line therapy for steroid-resistant aGvHD was rATG 2 mg/kg every other day for 2 to 4 doses. cGvHD was treated with prednisone or methylprednisolone with or without tacrolimus, while other agents were introduced if the response was incomplete.

### 2.2. Definition of Relapse

Relapse was diagnosed as the recurrence of primary disease, defined as blasts > 5% on BM biopsy or the presence of circulating blasts on peripheral blood (PB) samples. Persistence or recurrence of minimal residual disease (MRD) detected in PB or BM samples either by flow cytometry or next-generation sequencing (NGS) was not defined as relapse. The date of relapse was the date of the BM biopsy or diagnostic PB sample confirming relapse.

### 2.3. Definition of Graft-Versus-Host Disease (GvHD)

Acute GvHD was diagnosed, staged, and graded on at least a weekly basis through day 84 according to consensus criteria [19]. Chronic GvHD (cGvHD) was diagnosed, staged, and graded on at least a monthly basis after day 84 according to consensus criteria [20]. The transplant team performed the grading.

### 2.4. Statistical Analysis

Clinical outcomes included OS, event-free survival (EFS), NRM, relapse, aGvHD, and cGvHD. Continuous variables are presented as median and range, and categorical variables are summarized as counts and percentages. Survival data were estimated using the Kaplan–Meier curves for visualization and log-rank test for statistical analysis. Right-censoring at time of relapse was performed for the NRM analysis. Clinical endpoints of OS, EFS, NRM, and relapse were measured at 1 and 3 years after alloSCT. Statistical analyses after study enrollment were conducted via GraphPad Prism (Ver 9.1.0, GraphPad, San Diego, CA, USA) and SAS (Ver 9.4, SAS Institute Inc., Cary, NC, USA). For all analyses, the threshold for achievement of statistical significance was *p* < 0.05.

## 3. Results

### 3.1. Patient Characteristics

Patient and transplant characteristics are summarized in Table 1 and Table 2, with 24 and 22 subjects in the Flu/Bu4 and Flu/Bu4/TBI arms, respectively. The median age of the study population was 49 years (range: 22–63 years), and 58.7% were male. There were 34 AML patients, including four patients diagnosed with secondary AML, and all were in either first complete remission (CR1) or second complete remission (CR2) at the time of randomization. AML risk stratification was conducted using 2017 European LeukemiaNet (ELN) risk stratification [21]. Other patients were diagnosed with the following myeloid diseases: five patients had intermediate or high-risk myelodysplastic syndrome (MDS), three patients had myelofibrosis, three patients had chronic myelomonocytic leukemia, and one patient had chronic myeloid leukemia (CML) in myeloid blast crisis. The median follow-up after alloSCT until study termination was 27 months (range: 1–137 months).

### 3.2. Relapse

The 3-year cumulative incidence of relapse (CIR) of myeloid diseases (n = 46) was 47.4% in the Flu/Bu4 group and 38.9% in the Flu/Bu4/TBI group (HR = 0.83, 95% CI, 0.32 to 2.16, log-rank *p* = 0.70; Figure 2A). Disease relapse in the first year in patients with AML (n = 34) who received Flu/Bu4 was 33.3% versus 18.8% in those who received Flu/Bu4/TBI, and 44.4% versus 25.0%, respectively, over three years (HR = 0.58 [95% CI, 0.19 to 1.81], *p* = 0.35; Figure 2B). In a subgroup analysis of patients aged 50 years and younger (Flu/Bu4 n = 11; Flu/Bu4/TBI n = 9), the 3-year CIR was 30.8% in the Flu/Bu4 group versus 10.0% in the Flu/Bu4/TBI group (HR = 0.23 [95% CI, 0.05 to 1.61] log-rank *p* = 0.15; Figure 2C).

### 3.3. Survival

No graft failures were observed. In the overall cohort, OS, EFS, and relapse at 1 year were 73.9%, 63.0%, and 26.1%, respectively. Three-year OS was 37.5% in patients who received Flu/Bu4 and 50% in patients treated with Flu/Bu4/TBI (HR = 0.88, 95% CI, 0.40 to 1.92, log-rank *p* = 0.74; Figure 3A). Three-year EFS was 33.3% in the Flu/Bu4 group versus 50% in the Flu/Bu4TBI group (*p* = 0.70).

In a subgroup analysis of 34 subjects with AML, OS was 73.5% at 1 year. EFS at 1 year was numerically higher in AML patients who received Flu/Bu4 (77.8%) versus Flu/Bu4/TBI (68.8%) (*p* = 0.4). OS at 3 years was 44.4% in the Flu/Bu4 group versus 50% in the Flu/Bu4/TBI group (HR = 0.98, 95% CI, 0.39 to 2.50, log-rank *p* = 0.97; Figure 3B).

In a separate subgroup analysis of patients aged 50 years and younger (n = 20), 3-year OS was 70.0% in the Flu/Bu4/TBI group (n = 9) versus 46.2% in the Flu/Bu4 group (n = 11) (HR = 0.51, 95% CI, 0.15 to 1.84, log-rank *p* = 0.32; Figure 3C).

### 3.4. Non-Relapse Mortality

At day +100 from alloSCT, NRM was 2.2%, as one patient in the Flu/Bu4/TBI group died due to multi-organ failure. The 1-year NRM was 8.3% in the Flu/Bu4 cohort and 9.1% in the Flu/Bu4/TBI cohort. Among the 46 patients with myeloid disease, 12 patients (six patients in each cohort) died within one year of transplantation, including seven from disease relapse, two from aGvHD, one from infection, one from a fall, and one from organ failure (Table 3).

Among the 34 patients with AML, the 1-year NRM was 5.5% in the Flu/Bu4 group and 12.5% in the Flu/Bu4/TBI group. The 3-year NRM was 11.1% in the Flu/Bu4 cohort and 25.0% in the Flu/Bu4/TBI group (HR = 2.11, 95% CI 0.51 to 8.83, *p* = 0.65)

### 3.5. Development of GvHD

At day +100 from alloSCT, grade II–IV and III–IV aGvHD rates were 45.7% and 10.9%, respectively. The cumulative incidence of aGvHD at day +100 was numerically different in the Flu/Bu4 group and Flu/Bu4/TBI group (grade II–IV, 29.2% versus 63.6%, *p* < 0.1; grade III–IV, 8.3% versus 13.6, *p* = 0.7), though these findings did not reach statistical significance. At 1 year from alloSCT, limited cGvHD and extensive cGvHD rates were 26.1% and 13.0%, respectively. The cumulative incidence of cGvHD at 1 year was 29.2% and 50.0% in the Flu/Bu4 and Flu/Bu4/TBI groups, respectively (*p* = 0.5).

## 4. Discussion

Although a retrospective study found that Flu/Bu4/TBI conditioning prior to alloSCT was associated with improved overall survival when compared to Flu/Bu4 [19], those findings were not confirmed in this prospective randomized study. This prospective, randomized study’s results show that augmenting Flu/Bu4 with 400 cGy TBI resulted in a non-significant trend toward a lower risk of relapse but failed to provide a statistically significant benefit in OS.

Research has been undertaken to evaluate Flu/Bu4 as a MAC regimen in subjects with myeloid malignancies [8,23,24]. In a study of Flu/Bu4 as MAC for HLA-identical sibling alloSCT in AML and MDS, Flu/Bu4 was associated with a 4-year DFS of 54% and OS of 62%, though a 30% risk of relapse was also identified [24]. For AML patients with a high risk of relapse, intensifying the conditioning regimen may be an option, yet limitations of chemotherapy augmentation are pronounced. Andersson et al. performed a randomized phase III clinical trial testing the addition of clofarabine to the Flu/Bu4 regimen [25] and found that the observed relapse benefit of adding clofarabine was eroded by higher transplant-related mortality (TRM), resulting in no benefit in survival.

Instead of adding chemotherapeutics to intensify the conditioning regimen, additional TBI was studied. Deeg et al. showed that the addition of 200 cGy TBI to treosulfan/fludarabine reduced the incidence of relapse and conferred superior 6-month PFS [26]. In patients with AML, the patients receiving TBI 200 cGy had a 16% relapse incidence while the patients not given TBI had 35% relapse. However, final analysis showed that these differences were not statistically significant, even without increasing toxicity or NRM.

The present study, which evaluated the addition of 400 cGy to a myeloablative Flu/Bu4 conditioning regimen, also did not note an increase in PFS or OS. Though the incidence of disease relapse in the Flu/Bu4/TBI group trended lower than in the Flu/Bu4 group, this difference did not translate into better survival rates due to possible increases in NRM. Subgroup analysis of younger patients (aged ≤50 years) exhibited a trend toward improved OS and lower incidence of relapse with the Flu/Bu4/TBI regimen (Figure 2C). However, since the study was underpowered and unable to detect any statistically significant differences between the two groups in the subgroup analysis, we cannot preclude that younger patients with AML would benefit from Flu/Bu4/TBI conditioning.

Though low-dose TBI does not appear to increase the risk of subsequent malignancy [27,28], the addition of 400 cGy TBI in conditioning regimens may induce more tissue damage, which was observed in this study: the incidence of grade II–IV aGvHD was significantly higher with Flu/Bu4/TBI vs. Flu/Bu4, though comparisons of grade ≥III aGvHD and cGvHD did not reach statistical significance (Table 4).

A limitation is that our study data are from a single institute. The study was underpowered due to small sample sizes and the challenges of multiple comparisons. The conclusiveness of subgroup analyses was subsequently limited. Furthermore, both GvHD prophylaxis and second-line treatment for aGvHD differed from the modalities employed in current practice, which may have skewed the regimen-related toxicity.

The lack of busulfan pharmacokinetics is another limitation of this study. Although busulfan pharmacokinetics was not used for either arm of this study, the patients in the Flu/Bu4/TBI arm may have been at risk of excess NRM when cumulative busulfan exposure was higher. Additionally, literature shows that investigators aim to mitigate the potential for additive toxicity with higher busulfan exposure and TBI. Prior to the addition of 400 cGy TBI to their Flu/Bu4 regimen, Russell et al. initially targeted busulfan exposure of 4500 uM/min [29], while after the addition of TBI, Ousia et al. employed a lower target busulfan exposure of 3750 uM/min [30].

Another limitation in this study was that pre-transplant MRD status was not routinely included for the study participants, although this should have been offset by the randomization of the study population.

The BMT CTN 0901 trial identified pre-transplant MRD status as a predictive marker of post-transplant outcomes in patients who underwent RIC [8,9]. A growing body of evidence suggests that MRD positivity prior to alloSCT is an independent predictor of poor post-transplant outcomes in AML [31,32,33,34,35]. Therefore, we cannot completely rule out the possibility that pre-transplant MRD status might skew the results of this clinical study.

Though many studies have sought to identify an optimal conditioning regimen to reduce post-transplant relapse and to increase survival, the trials couldn’t provide an ideal conditioning regimen in allogeneic stem cell transplant. Since many agents targeting specific mutations are in development, post-transplant maintenance regimens may be the solution to achieve these goals instead of attempting to escalate the dose intensity of the conditioning regimen through the addition of further agents. If pre- or post-transplant mutation study identifies a driving mutation of the myeloid diseases, a targeted therapy may be used in the post-transplant consolidation period.

## 5. Conclusions

In conclusion, this clinical trial compared two conditioning regimens, Flu/Bu4 and Flu/Bu4/TBI, in patients with myeloid diseases who received a related or unrelated alloSCT. The addition of 400 cGy TBI to the Flu/Bu4 regimen did not show survival advantage compared to the Flu/Bu4 regimen.

## Figures and Tables

**Figure 1 cancers-17-01140-f001:**
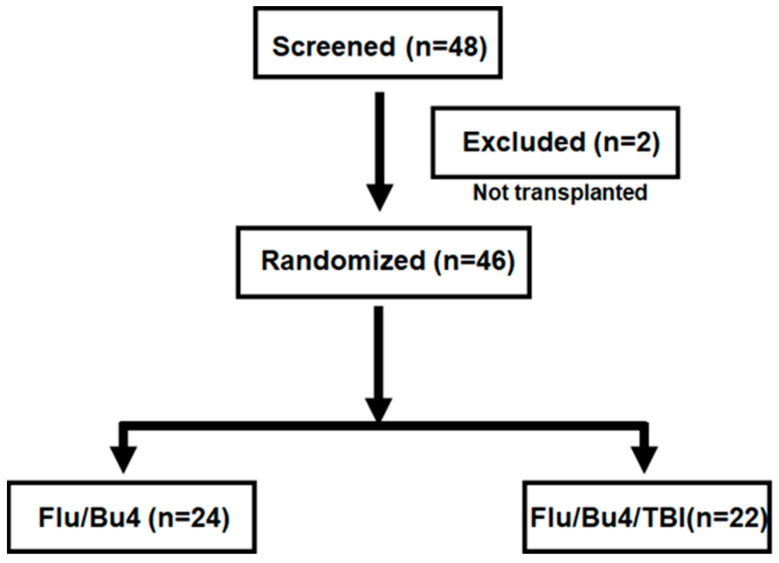
Consort diagram of patient recruitment and randomization.

**Figure 2 cancers-17-01140-f002:**
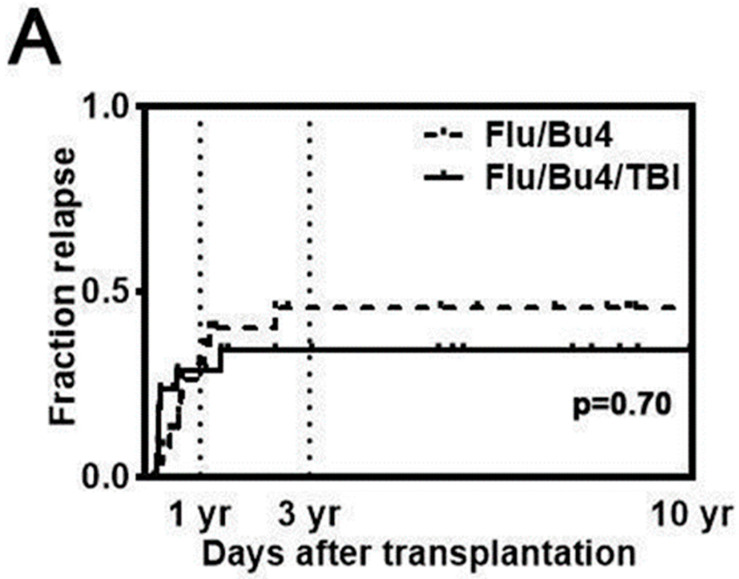
Relapse risk after alloSCT with Flu/Bu4 or Flu/Bu4/TBI conditioning. (**A**) Relapse risk among patients with myeloid diseases (n = 46); (**B**) relapse risk among patients with AML (n = 34); (**C**) relapse risk among patients 50 years old and younger (n = 23).

**Figure 3 cancers-17-01140-f003:**
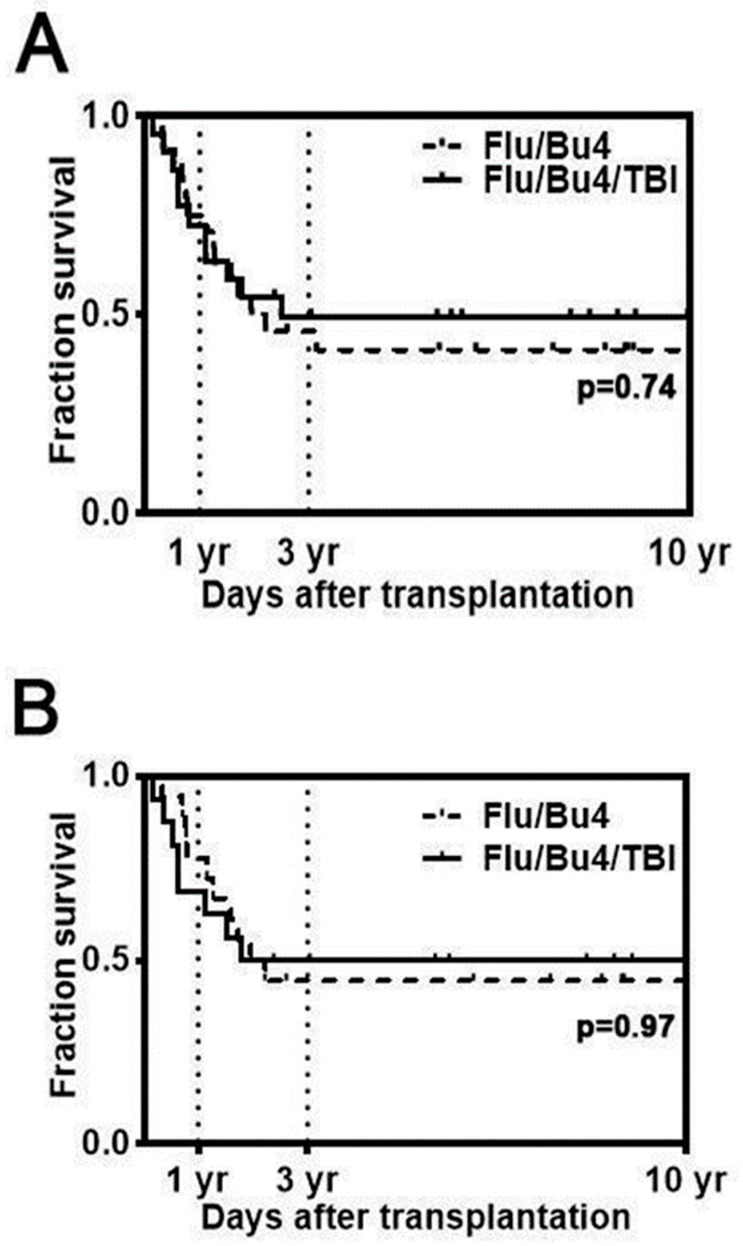
Overall survival after alloSCT with Flu/Bu4 or Flu/Bu4/TBI conditioning. (**A**) OS of patients with myeloid diseases (n = 46); (**B**) OS of patients with AML (n = 34); (**C**) OS of patients 50 years old and younger (n = 23).

**Table 1 cancers-17-01140-t001:** Comparison of patient and disease characteristics.

Variable	Flu/Bu4 (n = 24) Count (%)	Flu/Bu4/TBI (n = 22) Count (%)	*p*-Value
Age, years			0.91
Median (IQR)	49.0 (40.5–60.5)	53.0 (42.0–58.0)	
Range (Min–Max)	25.0–62.0	22.0–63.0	
Sex			0.96
Female	10 (41.7)	9 (40.9)	
Male	14 (58.3)	13 (59.1)	
HCT-CI ^1^			0.49
0–2	22 (91.7)	22 (100)	
≥3	2 (8.3)	0 (0)	
Diagnosis			0.89
AML	18 (75.0)	16 (72.7)	
Favorable risk	5 (27.8)	0 (0)	0.06
Intermediate risk	4 (22.2)	3 (18.8)	
Adverse risk	9 (50.0)	13 (81.2)	
CR status			1.00
CR1	13 (72.2)	11 (68.6)	
CR2	5 (27.8)	5 (31.2)	
MDS	3 (12.5)	2 (9.1)	
MF	1 (4.2)	2 (9.1)	
CML-BC	0 (0)	1 (4.6)	
CMML	2 (8.3)	1 (4.6)	

^1^ Pre-transplantation comorbidities were assessed by means of the Hematopoietic Stem Cell Transplant Comorbidity Index (HCT-CI) score [22].

**Table 2 cancers-17-01140-t002:** Comparison of donor and transplant characteristics.

Variable	Flu/Bu4 (n = 24) Count (%)	Flu/Bu4/TBI (n = 22) Count (%)	*p*-Value
Donor			0.23
Sibling, identical	9 (37.5)	4 (18.2)	
Unrelated			
HLA matched	13 (54.2)	13 (59.1)	
HLA mismatched	2 (8.3)	5 (22.7)	
Donor/Recipient Sex			0.20
Male to Male	10 (41.7)	9 (40.9)	
Male to Female	4 (16.7)	8 (36.3)	
Female to Male	4 (16.7)	4 (18.2)	
Female to Female	6 (25.0)	1 (4.6)	
Stem cell source			0.41
PBSCs	22 (91.7)	18 (81.8)	
BM	2 (8.3)	4 (18.2)	
GvHD prophylaxis			
Tac/MTX	24 (100)	22 (100)	
CMV status			0.37
Donor+ to Recipient+	8 (33.3)	6 (27.3)	
Donor+ to Recipient−	1 (4.2)	5 (22.7)	
Donor− to Recipient+	8 (33.3)	6 (27.3)	
Donor− to Recipient−	7 (29.2)	5 (22.7)	
Time to engraftment			
Neutrophil (days)			
Median (IQR)	13.0 (12.0–15.0)	13.0 (12.0–14.0)	
Range (Min–Max)	11.0–23.0	10.0–20.0	
Platelet (days)			
Median (IQR)	12.0 (10.0–15.0)	13.0 (12.0–18.0)	
Range (Min–Max)	10.0–21.0	9.0–20.0	

**Table 3 cancers-17-01140-t003:** Comparison of cause of death.

Variable	Flu/Bu4 Count (%)	Flu/Bu4/TBI Count (%)
Relapse	7 (53.8)	5 (45.4)
GvHD	1 (7.7)	2 (18.2)
Organ failure	1 (7.7)	2 (18.2)
Infection	2 (15.4)	2 (18.2)
Other cancer	1 (7.7)	0 (0)
Accident	1 (7.7)	0 (0)

**Table 4 cancers-17-01140-t004:** Comparison of incidence of GvHD.

Variable	Flu/Bu4 Count (%)	Flu/Bu4/TBI Count (%)	*p*-Value
Acute GvHD			
Grade II–IV	7 (29.2)	14 (63.6)	0.02
Grade ≥ III	2 (8.3)	3 (13.6)	0.66
Chronic GvHD			0.87
Limited	4 (16.7)	8 (36.4)	
Extensive	3 (12.5)	3 (13.6)	

## Data Availability

The data presented in this study are available on request from the corresponding author due to patient privacy.

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
