# Peer review of "Prospective, Randomized, Comparative Study of Myeloablative Fludarabine/Busulfan and Fludarabine/Busulfan/Total Body Irradiation Conditioning in Myeloid Diseases"

_cancers, 2025, doi:10.3390/cancers17071140_

Round 1
Reviewer 1 Report
Comments and Suggestions for Authors
General comments:
The article is well written and pleasant to read. Tables and graphics are clear, and references are relevant. Overall, the article is carefully prepared. The ethics statements are adequate.
The research question is relevant, and it is highly important that the different conditioning regimens are compared in prospective randomized fashion. The study groups were rather comparable in patients’ and disease characteristics, and excluding TBI, both groups received identical treatment. A significant weakness of the study is the low number of patients leading to lack of statistical power and reducing the reliability of the study findings. Furthermore, both gvhd prophylaxis and second line treatment for acute GVHD differed from current practice which may have increased the regimen-related toxicity. The study was opened already 15 years ago so supportive care has been improved also in other respects.
Spesific comments:
row 44 and 46: missing p-values
row 135: the MTX dose is lower than conventionally used, what is the background, and could this contribute to high incidence of gvhd (see comment on discussion)?
Statistical analysis:
Although the study was prematurely terminated it would be beneficial to provide the original calculation for sample size estimation.
3.4 Transplant-related mortality:
Can you state what proportion of patients that died due to toxicity were under/over 50 years of age.
Discussion:
Since the study was clearly underpowered to detect any statistically significant differences between the (sub)groups it does not preclude that younger (<50 years) AML patients would not benefit from Flu/Bu/TBI during the current era of improved management of gvhd and supportive care. This notion could be phrased in discussion.
Table1.
Can you provide the risk stratification used for AML patients in Method section.
Author Response
Thank you for your feedback and your time in reviewing our manuscript, Prospective, randomized, comparative study of myeloablative fludarabine/busulfan and fludarabine/busulfan/total body irradiation conditioning in myeloid diseases.
We found your comments to be insightful and have addressed each of them below. Thank you in advance for your consideration of these revisions.
Specific comments:
row 44 and 46: missing p-values:
- Thank you for your comment. Per your comment, we have added these p values on lines 44 and 46.
row 135: the MTX dose is lower than conventionally used, what is the background, and could this contribute to high incidence of gvhd (see comment on discussion)?
- Thank you for your comment. The dose of MTX used in this study, in combination with tacrolimus, has been tested in clinical trials for GvHD prophylaxis. One of the trials (Przepiorka et al, Blood, 1996) showed that 34% (95% CI, 17% to 52%) and 17% (95% CI, 3% to 31%) of patients who received MTX intravenously at 5 mg/m2 on days 1, 3, 6, and 11 experienced Grades 2-4 GvHD and Grades 3-4 GvHD, respectively. Overall survival at 1 year among these patients was 47% (95% CI, 26% to 66%).
- This trial, as well as other studies (Devine et al, Biol Blood Marrow Transplant, 1997; Przepiorka et al, Bone Marrow Transplant, 1999), did not observe higher rates of GvHD with mini-methotrexate (ie, MTX 5 mg/m2). At our cancer center, we currently use this mini-methotrexate regimen if the patient requires tacrolimus/MTX for GvHD prophylaxis.
Statistical analysis:
Although the study was prematurely terminated it would be beneficial to provide the original calculation for sample size estimation.
- Thank you for your comment. Please find below, the original null hypothesis and sample size calculation that was included within the IRB-approved study protocol:
- The null hypothesis is that the 1 year relapse rate is 34%. It is the goal of this study to show that when Flu/Bu4/TBI is used, the relapse rate decreases to 13.6% vs a relapse rate of 40% with the Flu/Bu4 regimen. Thus, we will evaluate the hypotheses:
- HO: θA ≤ θB
- HA: θA > θB
- where θA= rate of relapse in Arm A (FluBu) which is assumed to be 0.34, θB= rate of relapse in Arm B (FluBuTBI)
- Due to the lack of homogeneity in the outcome as a result of risk status (low versus high), it will be useful to stratify on risk status. Thus, a one-sided log-rank test stratified on risk status will be conducted to ascertain if Flu/Bu4/TBI reduces the relapse rate versus Flu/Bu4 using a 5% level of significance.
- Accrual, Registration, and Follow Up:
- The targeted sample size is 54 subjects. It is estimated that four years of accrual will be necessary to enroll this number of subjects. About 50 patients (49-52) underwent allogeneic transplantation at this center, indicating that an appropriate number of subjects will be available to complete this study in the time frame anticipated. Subjects will enroll uniformly over the accrual period. The randomization will be stratified by relapse risk status (high versus low). Low relapse will include: AML CR1 and CML chronic phase. All other AML, MPD and MDS patients will be considered high relapse risk. An AML CR1 patient with a history of MDS will be considered high relapse risk. All patients who are Flt3/ITD positive at any time will be considered high relapse risk. Upon establishing eligibility, subjects will be randomized in equal numbers to the Flu/Bu4 and Flu/Bu4/TBI arms using permuted blocks within strata.
- All subjects will be followed for one year from time of transplantation, during which they will be monitored for the effects of treatment through regular clinic visits. Additional follow-up data beyond one year of progression-free survival will be obtained from HUMC Cancer Center Database.
- Sample Size and Power Calculations
- The main objective is to compare the relapse-free surviving proportions at 1 year in myeloid malignancy patients who are randomized to either Flu/Bu4/TBI or Flu/Bu4 conditioning regimen.
- In examining the time to relapse, we consider an analysis of the proportion surviving without relapse/progression at the 1 year endpoint. A comparison of this event between the two treatment arms will be conducted using log-rank test statistics at the 5% level of significance. Based on this statistic, the calculation of sample size was obtained using PASS 2008. According to the results from previous studies [De Lima et al], the 1-year relapse rate in patients with myeloid malignancies being treated with Flu/Bu4 is 34%. We posit that when the Flu/Bu4/TBI regimen is used the relapse rate will be reduced to 13.6%, which is 40% of the relapse risk in the Flu/Bu4 regimen arm. Then, a one-sided log-rank test will be used to examine evidence for a reduction in disease relapse. For the nominal power of 80%, significance level of 5%, 4 year accrual, 1 year follow-up and 10% drop out rate in both Flu/Bu4/TBI and the Flu/Bu4 treatment arm, 52 patients (26 per arm) will be required to detect the 20.4 difference (60% reduction in relapse rate). This calculation achieved 80.06%. To adjust for balanced samples within the risk status, the study will enroll a maximum of 54 patients (27 per arm).
- The null hypothesis is that the 1 year relapse rate is 34%. It is the goal of this study to show that when Flu/Bu4/TBI is used, the relapse rate decreases to 13.6% vs a relapse rate of 40% with the Flu/Bu4 regimen. Thus, we will evaluate the hypotheses:
3.4 Transplant-related mortality:
Can you state what proportion of patients that died due to toxicity were under/over 50 years of age.
Thank you for your comment. Please see below for the data requested:
- Flu/Bu4 group, age under 50 (7 pts): relapse 4, GvHD 1, infection 1, fall (trauma) 1
- Flu/Bu4 group, age over 50 (6 pts): relapse 3, GvHD 0, infection 1, organ failure 1, other cancer 1
- Flu/Bu4/TBI group, age under 50 (3 pts): relapse 1, infection 1, organ failure 1
- Flu/Bu4/TBI group, age over 50 (8 pts): relapse 4, GvHD 2, infection 1, organ failure 1
Discussion:
Since the study was clearly underpowered to detect any statistically significant differences between the (sub)groups it does not preclude that younger (<50 years) AML patients would not benefit from Flu/Bu/TBI during the current era of improved management of gvhd and supportive care. This notion could be phrased in discussion.
- Thank you for your comment. We agree that the underpowered nature of the study limits the conclusiveness of subgroup analyses, and have incorporated this notion into lines 298-302.
Table1.
Can you provide the risk stratification used for AML patients in Method section.
- Thank you for your comment. The risk stratification approach used for AML patients was based on the 2017 European LeukemiaNet construct. We have added this on lines 197-198.

Reviewer 2 Report
Comments and Suggestions for Authors
Overall, this is an important addition to the literature given that it is a rare prospective study regarding conditioning for allo-HCT. There are certainly limitations in drawing any firm conclusions from the study given that it was terminated early and is likely severely underpowered to detect any difference in transplant-related and survival outcomes. Nevertheless, the HCT community will benefit from having these data available.
Comments:
-Although the study was terminated early: Please reference the power calculations and how they informed the planned enrolment of the study. What was the primary endpoint?, the expected difference in primary endpoint between the two arms?, and the number of patients that were to be enrolled to ensure the study was appropriately powered?
-Please clarify the flow of patients through the study (perhaps with a consort-type diagram) as the current description is confusing (48 patients were enrolled, but later it is noted that the study was halted after 52 enrolled, and finally 46 patients are included in the analysis presented here).
-The lack of Bu PK should be acknowledged as a weakness of the study: Although BU PK was not used for either arm of the study, it may be important to point out that those receiving TBI may have been at risk of excess NRM when Bu exposure was high given the possibility of additive toxicity of high Bu exposure + TBI. Prior to the addition of 4 Gy TBI to their FluBu protocol, the Calgary group initially targeted a BU exposure of 4500 uM/min (Russell et al. BBMT 2013 Sep;19(9):1381-6. doi: 10.1016/j.bbmt.2013.07.002). However, after the addition of TBI a lower BU exposure of 3750 uM/min was targeted (Ousia et al. Clinical Transplantation 2020 Sep;34(9):e14018. doi: 10.1111/ctr.14018).
-Please provide more details on TRM/NRM: In the abstract, NRM is noted to be 33.3% vs. 20% (over what time frame?), however these numbers are not presented in the results section. In fact, the 1-year TRM as presented in the results section are similar between the two groups (8.3 vs 9.1%).
-The conclusion that 400 cGY TBI "did not improve OS due to higher regimen-related mortality" is too confident a statement given that the study was likely severely underpowered to detect significant differences in any outcome.
-Are there any groups of patients that the authors think might benefit from the addition of TBI? This should be added to the discussion for hypothesis generation.
Minor:
- Specify the formulation rATG used (? Thymoglobulin ? Grafalon)
- How was AML-risk classified (table 1)? ELN? Other risk stratification system?
- Please define event-free survival (what were the events?)
- A small discussion of possible late toxicity of adding 4 Gy TBI would be beneficial in the discussion: From a subsequent malignancy standpoint, the addition of "low dose" TBI does not appear to be harmful (Nunez. Blood Advances 2022 Feb 8;6(3):767-773. & Baker. Blood. 2019 Jun 27;133(26):2790-2799).
Author Response
Thank you for your feedback and your time in reviewing our manuscript, Prospective, randomized, comparative study of myeloablative fludarabine/busulfan and fludarabine/busulfan/total body irradiation conditioning in myeloid diseases.
We found your comments to be insightful and have addressed each of them below. Thank you in advance for your consideration of these revisions.
Comments:
-Although the study was terminated early: Please reference the power calculations and how they informed the planned enrolment of the study.
- Thank you for your comment. Please find below, the original null hypothesis and sample size calculation that was included within the IRB-approved study protocol:
- The null hypothesis is that the 1 year relapse rate is 34%. It is the goal of this study to show that when Flu/Bu4/TBI is used, the relapse rate decreases to 13.6% vs a relapse rate of 40% with the Flu/Bu4 regimen. Thus, we will evaluate the hypotheses:
- HO: θA ≤ θB
- HA: θA > θB
- where θA= rate of relapse in Arm A (FluBu) which is assumed to be 0.34, θB= rate of relapse in Arm B (FluBuTBI)
- Due to the lack of homogeneity in the outcome as a result of risk status (low versus high), it will be useful to stratify on risk status. Thus, a one-sided log-rank test stratified on risk status will be conducted to ascertain if Flu/Bu4/TBI reduces the relapse rate versus Flu/Bu4 using a 5% level of significance.
- Accrual, Registration, and Follow Up:
- The targeted sample size is 54 subjects. It is estimated that four years of accrual will be necessary to enroll this number of subjects. About 50 patients (49-52) underwent allogeneic transplantation at this center, indicating that an appropriate number of subjects will be available to complete this study in the time frame anticipated. Subjects will enroll uniformly over the accrual period. The randomization will be stratified by relapse risk status (high versus low). Low relapse will include: AML CR1 and CML chronic phase. All other AML, MPD and MDS patients will be considered high relapse risk. An AML CR1 patient with a history of MDS will be considered high relapse risk. All patients who are Flt3/ITD positive at any time will be considered high relapse risk. Upon establishing eligibility, subjects will be randomized in equal numbers to the Flu/Bu4 and Flu/Bu4/TBI arms using permuted blocks within strata.
- All subjects will be followed for one year from time of transplantation, during which they will be monitored for the effects of treatment through regular clinic visits. Additional follow-up data beyond one year of progression-free survival will be obtained from HUMC Cancer Center Database.
- Sample Size and Power Calculations
- The main objective is to compare the relapse-free surviving proportions at 1 year in myeloid malignancy patients who are randomized to either Flu/Bu4/TBI or Flu/Bu4 conditioning regimen.
- In examining the time to relapse, we consider an analysis of the proportion surviving without relapse/progression at the 1 year endpoint. A comparison of this event between the two treatment arms will be conducted using log-rank test statistics at the 5% level of significance. Based on this statistic, the calculation of sample size was obtained using PASS 2008. According to the results from previous studies [De Lima et al], the 1-year relapse rate in patients with myeloid malignancies being treated with Flu/Bu4 is 34%. We posit that when the Flu/Bu4/TBI regimen is used the relapse rate will be reduced to 13.6%, which is 40% of the relapse risk in the Flu/Bu4 regimen arm. Then, a one-sided log-rank test will be used to examine evidence for a reduction in disease relapse. For the nominal power of 80%, significance level of 5%, 4 year accrual, 1 year follow-up and 10% drop out rate in both Flu/Bu4/TBI and the Flu/Bu4 treatment arm, 52 patients (26 per arm) will be required to detect the 20.4 difference (60% reduction in relapse rate). This calculation achieved 80.06%. To adjust for balanced samples within the risk status, the study will enroll a maximum of 54 patients (27 per arm).
- The null hypothesis is that the 1 year relapse rate is 34%. It is the goal of this study to show that when Flu/Bu4/TBI is used, the relapse rate decreases to 13.6% vs a relapse rate of 40% with the Flu/Bu4 regimen. Thus, we will evaluate the hypotheses:
What was the primary endpoint?, the expected difference in primary endpoint between the two arms?, and the number of patients that were to be enrolled to ensure the study was appropriately powered?
- Thank you for your comment. The primary objective of this single institution randomized Phase III trial is to compare the 1-year relapse rate of myeloid malignancy in patients receiving two different conditioning regimens. In this study, subjects will be randomly assigned to one of 2 conditioning regimens: fludarabine and busulfan or fludarabine, busulfan and low-dose total body irradiation. The myeloid malignancies include acute myeloid leukemia (AML), chronic myeloid leukemia (CML) and other myeloproliferative diseases, and myelodysplastic syndromes (MDS).
- According to the results from previous studies [De Lima et al], the 1-year relapse rate in patients with myeloid malignancies being treated with Flu/Bu4 is 34%. We posit that when the Flu/Bu4/TBI regimen is used the relapse rate will be reduced to 13.6%, which is 40% of the relapse risk in the Flu/Bu4 regimen arm. Then, a one-sided log-rank test will be used to examine evidence for a reduction in disease relapse. For the nominal power of 80%, significance level of 5%, 4 year accrual, 1 year follow-up and 10% drop out rate in both Flu/Bu4/TBI and the Flu/Bu4 treatment arm, 52 patients (26 per arm) will be required to detect the 20.4 difference (60% reduction in relapse rate). This calculation achieved 80.06%. To adjust for balanced samples within the risk status, the study will enroll a maximum of 54 patients (27 per arm).
-Please clarify the flow of patients through the study (perhaps with a consort-type diagram) as the current description is confusing (48 patients were enrolled, but later it is noted that the study was halted after 52 enrolled, and finally 46 patients are included in the analysis presented here).
- Thank you for your comment. In accordance with your feedback, we have adapted the description in lines 95-99 to clarify enrollment, and added Figure 1, a consort diagram depicting patient recruitment and randomization.
-The lack of Bu PK should be acknowledged as a weakness of the study: Although BU PK was not used for either arm of the study, it may be important to point out that those receiving TBI may have been at risk of excess NRM when Bu exposure was high given the possibility of additive toxicity of high Bu exposure + TBI. Prior to the addition of 4 Gy TBI to their FluBu protocol, the Calgary group initially targeted a BU exposure of 4500 uM/min (Russell et al. BBMT 2013 Sep;19(9):1381-6. doi: 10.1016/j.bbmt.2013.07.002). However, after the addition of TBI a lower BU exposure of 3750 uM/min was targeted (Ousia et al. Clinical Transplantation 2020 Sep;34(9):e14018. doi: 10.1111/ctr.14018).
- Thank you for your comment. We agree that the lack of busulfan pharmacokinetics was a key limitation of this study. We have added a comment about this limitation to the discussion section (lines 317-324), and incorporated the references you kindly shared.
-Please provide more details on TRM/NRM: In the abstract, NRM is noted to be 33.3% vs. 20% (over what time frame?), however these numbers are not presented in the results section. In fact, the 1-year TRM as presented in the results section are similar between the two groups (8.3 vs 9.1%).
- Thank you for your comment. We recognize that these can be confusing, and have adjusted nomenclature and statistics to NRM globally throughout the manuscript, as this would be the better description for this study. These changes can be found in the abstract (lines 44-46), section 3.4 (lines 248-257).
-The conclusion that 400 cGY TBI "did not improve OS due to higher regimen-related mortality" is too confident a statement given that the study was likely severely underpowered to detect significant differences in any outcome.
- Thank you for your comment. We agree that the lack of statistical power in this study limits our ability to draw clear conclusions, and have softened this statement in lines 272-276.
-Are there any groups of patients that the authors think might benefit from the addition of TBI? This should be added to the discussion for hypothesis generation.
- Thank you for your comment. We believe that the trends observed in this study among young patients, including improved OS and a lower risk of relapse, merit further investigation of Flu/Bu4/TBI conditioning in this sub-population. We have added this in lines 298-99, and juxtaposed that information alongside the impact of the limited statistical power of this study.
Minor:
Specify the formulation rATG used (? Thymoglobulin ? Grafalon)
- Thank you for your comment. We have added the specific rATG formulation used (Thymboglobulin) in line 121.
How was AML-risk classified (table 1)? ELN? Other risk stratification system?
- Thank you for your comment. The risk stratification approach used for AML patients was based on the 2017 European LeukemiaNet construct. We have added this on lines 197-198.
Please define event-free survival (what were the events?)
- Thank you for your comment. The events are defined in Table 3.
A small discussion of possible late toxicity of adding 4 Gy TBI would be beneficial in the discussion: From a subsequent malignancy standpoint, the addition of "low dose" TBI does not appear to be harmful (Nunez. Blood Advances 2022 Feb 8;6(3):767-773. & Baker. Blood. 2019 Jun 27;133(26):2790-2799).
- Thank you for your comment. In accordance with your feedback, we have addressed that low-dose TBI does not appear to increase the risk of subsequent malignancy on lines 303-304, and used the references you kindly provided.

Reviewer 3 Report
Comments and Suggestions for Authors
Suh et al. presented the findings of a prospective trial that compared two myeloablative conditioning regimens, fludarabine/busulfan versus fludarabine/busulfan/total body irradiation, in the context of myeloid diseases. The conclusion drawn was that the inclusion of 400 cGy TBI to the Flu/Bu4 regimen did not confer a survival benefit when compared to the Flu/Bu4 regimen alone. The manuscript is exceptionally well composed. Aside from the small sample sizes, a significant limitation of this study is the lack of information regarding the pre-transplant MRD status.
I have a few suggestions for the authors to consider:
Main comments:
- Despite the limited sample size, it could be worthy to analyze the cytogenetic groups (Favorable/Intermediate versus Adverse) at diagnosis to determine if the additional 4 Gy of TBI affects the relapse rate. Could a similar analysis be conducted comparing CR1 and CR2?
- If there are cytogenetic data available prior to HCT, they could serve as an MRD marker: comparing normalized karyotype VS abnormal karyotype and analyzing the respective subgroups.
- The TRM appears to be favorable given the two myeloablative conditioning regimens employed in this study. However, rather than performing a subgroup analysis of patients aged 50 years and younger, perhaps you could consider conducting an OS subgroup analysis based on HCT-CI (0-1 versus higher or 0 versus higher)?
Some minor suggestions:
- Could you enlarge the graphs for better visibility?
- Regarding the definition of relapse, clarifying the cutoff percentage of blasts that indicate a relapse might be helpful.
- As for the definition of Graft-versus-Host Disease, I believe that treatment options for GVHD do not necessarily need to be included in this section.
Author Response
Thank you for your feedback and your time in reviewing our manuscript, Prospective, randomized, comparative study of myeloablative fludarabine/busulfan and fludarabine/busulfan/total body irradiation conditioning in myeloid diseases.
We found your comments to be insightful and have addressed each of them below. Thank you in advance for your consideration of these revisions.
I have a few suggestions for the authors to consider:
Main comments:
Despite the limited sample size, it could be worthy to analyze the cytogenetic groups (Favorable/Intermediate versus Adverse) at diagnosis to determine if the additional 4 Gy of TBI affects the relapse rate. Could a similar analysis be conducted comparing CR1 and CR2?
- Thank you for your comment. We agree that additional analyses to further establish the cohorts who may benefit from the addition of TBI would be valuable. However, during our previous attempts to run univariate and multivariate analyses based on clinical characteristics including risk category, and CR1 vs CR2, the sample sizes were too small to observe statistical differences.
If there are cytogenetic data available prior to HCT, they could serve as an MRD marker: comparing normalized karyotype VS abnormal karyotype and analyzing the respective subgroups.
- Thank you for your comment. We agree that cytogenetic data prior to HCT may serve as an MRD marker, but unfortunately, not every patient had cytogenetic data both prior and 3 months after HCT, limiting the feasibility of this analysis.
The TRM appears to be favorable given the two myeloablative conditioning regimens employed in this study. However, rather than performing a subgroup analysis of patients aged 50 years and younger, perhaps you could consider conducting an OS subgroup analysis based on HCT-CI (0-1 versus higher or 0 versus higher)?
- Thank you for your comment. As above, we agree that additional analyses to further establish the cohorts who may benefit from the addition of TBI would be valuable. However, there was a substantial concentration to HCT-CI 1-2 (n=39/44). Only two patients in the Flu/Bu4 group had an HCT-CI score of 3. Therefore, we were unable to see statistical differences between HCT-CI score 1 and higher.
Some minor suggestions:
Could you enlarge the graphs for better visibility?
- Thank you for your comment. In accordance with your feedback, we have enlarged the graphs in Figure 2 (previously Figure 1) and Figure 3 (previously Figure 2).
Regarding the definition of relapse, clarifying the cutoff percentage of blasts that indicate a relapse might be helpful.
- Thank you for your comment. In accordance with your feedback, we have more clearly defined relapse in section 2.2 (lines 160-162) as blasts >5% on BM biopsy or the presence of circulating blasts on peripheral blood samples.
As for the definition of Graft-versus-Host Disease, I believe that treatment options for GVHD do not necessarily need to be included in this section.
- Thank you for your comment. In accordance with your feedback, we have moved the GvHD treatment content from section 2.3 to section 2.1 (Treatment). These edits can be found on lines 153-157.
